# Study on the Technology and Properties of Green Laser Sintering Nano-Copper Paste Ink

**DOI:** 10.3390/nano14171426

**Published:** 2024-08-31

**Authors:** Pengkun Li, Zilin Tang, Kaibo Guo, Guifeng Luo, Xihuai Wang, Shengbin Zhao, Mingdi Wang

**Affiliations:** School of Mechanical and Electrical Engineering, Soochow University, Suzhou 215000, China; pkli@stu.suda.edu.cn (P.L.); 20215229056@stu.suda.edu.cn (Z.T.); kaiboguohust@gmail.com (K.G.); 20225229068@stu.suda.edu.cn (G.L.); 20235229101@stu.suda.edu.cn (X.W.)

**Keywords:** nano-copper paste ink, green laser sintering, electronic circuits, resistivity

## Abstract

With the rapid development of integrated circuits, glass substrates are frequently utilized for prototyping various functional electronic circuits due to their superior stability, transparency, and signal integrity. In this experiment, copper wire was printed on a glass substrate using inkjet printing, and the electronic circuit was sintered through laser irradiation with a 532 nm continuous green laser. The relationship between resistivity and microstructure was analyzed after laser sintering at different intensities, scanning speeds, and iterations. The experimental results indicate that the conductivity of the sintered lines initially increases and then decreases with an increase in laser power and scanning speed. At the same power level, multiple sintering runs at a lower scanning speed pose a risk of increased porosity leading to reduced conductivity. Conversely, when the scanning speed exceeds the optimal sintering speed, multiple sintering runs have minimal impact on porosity and conductivity without altering the power.

## 1. Introduction

In recent years, the rapid development of the electronics industry has led to a continuous increase in the integration of electronic devices. Electronic components are evolving towards precision, intelligence, and low cost [1]. These components are widely utilized in flexible wearables, wireless communications, displays, microsensors, and other devices. Glass, as the electrode substrate, offers improved stability, transmission, and signal integrity. It is currently extensively used in electronic components and can also be applied to optical devices, 5G wireless communication, microfluidic chips, display devices, and other fields [2,3,4]. Therefore, there is growing interest in researching electrode preparation on its surface. Common methods include vapor deposition, lithography, and sputtering [5,6,7]. However, these processes involve complex preliminary procedures and high equipment costs. In contrast, inkjet-printed circuits have gained attention over the past decade due to their numerous advantages in terms of process and manufacturing costs compared to other deposition techniques.

When fabricating circuits, the use of suitable conductive metal inks is crucial [8]. Noble metals such as silver and gold are commonly employed as functional particles in conductive inks due to their high conductivity and stability in air [9,10,11]. However, these metals are prohibitively expensive for large-scale applications. Copper presents an appealing alternative due to its relatively high bulk conductivity and lower cost compared to precious metals [12]. The heat treatment process plays a critical role in the electrical conductivity of printing ink and the prevention of damage to substrate materials. Traditional heat treatment methods include oven-hot sintering, electric sintering, microwave sintering, and so on [13,14]. While hot sintering is slow and can cause damage to the substrate, it is not suitable for low-melting point substrate materials. Although electro-sintering is selective and effective, it is a continuous and slow process with low efficiency. Microwave sintering offers high selectivity but requires complex heating equipment with special design at a high cost, thus making it more suitable for preparing oxides. In recent years, laser sintering has become an important research focus for curing nano-conductive inks due to its small sintering point, high strength of sintered material, fast heating and cooling speed, as well as narrow heat-affected zone [15,16,17].

This study investigates the feasibility of using laser-sintered nanometer copper paste to form electronic circuits by utilizing a 532 nm continuous green laser to sinter 40 um-wide nanometer copper ink printed on a glass substrate.

## 2. Materials and Methods

### 2.1. Materials

In this paper, nano-copper paste is prepared by mixing nano-copper powder with organic solvent and additive, in which the mass fraction of copper powder is 40 wt.%. Add polyvinylpyrrolidone (PVP) solid powder into the beaker, add the appropriate amount of anhydrous ethanol into the beater, and stir it on a magnetic mixer for 3 min to dissolve it fully. The dissolved solution and nano-copper powder are mixed into the centrifugal tube according to the mass ratio of 6:4, and the mixed slurry is placed in the ultrasonic cleaning tank for ultrasonic dispersion for 30 min so that the nano-copper powder is evenly dispersed in the solution, that is, the evenly mixed nano-copper slurry is obtained. In this experiment, the diameter of nano-copper particles is 60–100 nm (average particle size 80 nm), and the average melting point temperature given by the manufacturer is 367 °C.

The viscosity and surface tension of nano-copper paste are two important physical properties that have an important effect on the application performance of nano-copper paste. As for the viscosity of the nano-copper paste, 3 M tape was pasted on the surface of the copper film, and the tape was quickly pulled apart after compression, as shown in Figure 1a. The adhesion of the nano-copper paste to the substrate was detected by observing the spalling of the coating in the grid. After a 3 M tape test, the spalling area of the sample was 0.358%, and the adhesion between the copper paste and the substrate reached 4B standard and met the application requirements. We used the contact angle to represent the surface tension of the nano-copper paste. The measurement results showed that, as shown in Figure 1b,c, the contact angle between the nano-copper paste and the substrate was 59° when PVP was not added; after PVP was added, the contact angle between the nano-copper paste and the substrate was 44°, the contact area between the nano-copper paste and the plane increased, and the contact angle decreased. Nano-copper paste can better wet the substrate.

### 2.2. Preparation of Samples

Inkjet printing was used to continuously deposit copper nano-ink onto a glass substrate, resulting in the formation of copper paste lines measuring 2 cm × 250 μm × 40 μm. Subsequently, a three-dimensional digital microscope was employed to perform contour scanning of the printed ink lines, and the scanning image is depicted in Figure 2. As a printing ink, copper nanoparticles should have good dispersion in the solvent, and their adhesion should meet the 4B standard. JETLAB 4, a nanomaterial deposition inkjet printing system from MICROFAB Technologies, Plano, TX, USA, was used.

### 2.3. Experimental Method

In this study, the sintering of nano-copper paste is carried out in the air. In order to prevent the oxidation of nano-copper, PVP is added to the nano-copper paste. PVP can not only be used as a dispersant but also be coated on nano-copper particles to prevent the oxidation of nano-copper particles. These decomposition products have a certain reducibility in sintering and can be nano-copper oxides such as cuprous oxide and copper oxide reduced to copper. During the experiment, the prepared sample was installed on the X-Y translation platform, a 532 nm continuous green laser with the maximum output power of 15 W was used as the light source, the CCD camera was connected to the computer, and the protection gas was turned on to observe the sintering process of the laser in the experiment. Different laser parameters (laser power density, scanning speed, scanning times) were used to carry out the sintering experiments, and the effects of laser power, scanning speed, and sintering times on the porosity and conductivity of sintering lines were compared. The schematic diagram of the experimental device is shown in Figure 3. After focusing, the spot diameter is 50 μm, and the laser scanning speed is controlled by the X-Y mobile station.

### 2.4. Surface Characterization

Surface morphology analysis of the copper wire after laser sintering was conducted using a scanning electron microscope (ZEISS EVO25, Jena, Germany). The main parameters of SEM selection were EHT = 20 KV, WD = 8.00 mm, Mag = 1.92 KX, and Signal A = SE1.

Three-dimensional topography (VHX-7000) measurement of inkjet-printed nano-copper wires was performed using a 3D digital microscope to inspect the width and thickness of the printed lines.

The XRD test was carried out with an X-ray diffractometer (D8 DISCOVER PLUS, Karlsruhe, Germany/BRUKER) to analyze the phase of the sintered line surface.

The resistance of the copper wire was measured using a digital multimeter and a four-point probe measurement device. The electrical conductivity, σ, was calculated by taking into account the thickness and length of the continuous line, as shown in the following equation:(1)$$\sigma =\frac{L}{R\times S}$$
where R represents the measured resistance, L denotes the length of the measured line, and S indicates the cross-sectional area of the copper wire.

The porosity of SEM images of laser-sintered lines was measured using ImageJ software. The specific operations are as follows: The Set Scale function in ImageJ software is utilized to measure and calibrate the scale on the scanning electron microscope image. Subsequently, the RGB image is converted to a 32-bit image for easier differentiation of color variations. A rectangular box is then used to select the range for calculation, and the threshold function is employed to adjust the threshold value of the box selection area until all pores in the figure are marked out. The calculated pore area is denoted as S1, while the area of the entire box selection area is denoted as S. The porosity P can then be obtained using the following formula:(2)$$P=\frac{S1}{S}$$

## 3. Results and Discussion

### 3.1. The Influence of Different Laser Parameters on the Sintering Process of the Lines

#### 3.1.1. Influence of Laser Power on Sintering Lines

The coated sample was placed on an X-Y mobile platform and exposed to the light spot of a 532 nm continuous green laser. The platform’s moving speed was set to 50 mm/s, ensuring that it moved only once. After each sintering process was completed, the laser power was adjusted to sinter the sample at different power levels of 0.5 W, 1 W, 1.5 W, 2 W, 2.5 W, and 3 W, respectively. The microscopic morphologies of the sintered lines under different laser powers are shown in Figure 4. In Figure 4a, the sintered surface morphology when the laser power was at 0.5 W is displayed. After laser irradiation, the nanoparticles on the copper paste layer surface are tightly arranged with smaller particles condensing around larger particles due to the low sintering temperature at this power level. At a laser power of 0.5 W, polymer additives are removed and nanoparticles (NPs) are in close contact with fine black regions at interfaces representing pores formed by particle arrangement during sintering. When the laser power is increased to 1 W, as shown in Figure 4b, the particles are arranged more closely and sintering necks begin to form between smaller-diameter copper nanoparticles due to greater driving force for sintering provided by an increase in laser power. Although sintering occurs at this laser power, the results are suboptimal, as many pores are still present. As the power increases to 1.5 W and 2 W, the pores at the interfaces gradually decrease and disappear. Plastically deformed copper nanoparticles fill the voids, and sintering necks form continuous veins that intertwine to create a network-like interconnected structure. This is attributed to the increase in sintering line surface temperature with laser power, which promotes Oswald ripening and further growth of the necks formed between nanoparticles, enhancing densification (see Figure 4c,d). When the laser power is further increased to 2.5 W and 3 W, the copper nanoparticles have sintered into blocks that are fully fused together, rendering the original nanoparticles no longer visible. The resulting surface exhibits a loose network structure with large pores, indicating decreased densification. At this point, it is possible that the sintering temperature may be far above the melting point of copper nanoparticles. In comparison to earlier experimental figures, it is evident that rapidly fused and agglomerated copper nanoparticles form much larger blocks within a short time (as shown in Figure 4e,f).

The microscopic morphology of sintered lines is a critical factor influencing their electrical conductivity [18]. In order to further illustrate the relationship between the sintered surface morphology of copper paste and its electrical properties, ImageJ software was used to process the SEM images of the sintered samples. Based on the differences in brightness and darkness within the images, pores were filled with red pixels. The processed images are shown in Figure 5.

Using ImageJ1.54j software, the porosity of the processed images was calculated to determine the area percentage of pores in each image. The resistivity values of the sintered lines under different laser powers were measured at room temperature using the four-point probe method, with three measurements taken for each sample and averaged. Subsequently, conductivity was calculated using Equation (1). The relationships between porosity, conductivity, and laser power are illustrated in Figure 6.

Figure 6 illustrates the relationship between the porosity and conductivity of nano-copper paste sintered lines with varying laser powers at a fixed laser scanning speed of 50 mm/s and at ambient temperature. An analysis of the data in the figure reveals that the conductivity of the sintered lines shows a trend of initially increasing and then decreasing with an increase in laser power. Microscopically analyzing the SEM images, this trend can be attributed to the fact that when the laser power is low, the sintering temperature is insufficient to drive the interconnection of nanoparticles, leading to a lack of direct contact between conductive phase particles. As a result, there are nano- or even micro-scale distances between particles, making it difficult to form conductive pathways. In this scenario, the conductivity primarily relies on the tunnel effect where only a small fraction of charged particles can traverse barriers while the rest are reflected, resulting in low conductivity. As power increases, a noticeable neck growth phenomenon is observed between copper nanoparticles with necks filling inter-particle pores establishing a percolation network for electron flow [19]. When the laser power reaches 2 W, the conductivity of the sintered line peaks at 3.46 × 10^6^ S/m, and the porosity on the sintered line surface reaches its minimum value of 11.15%, as shown in Figure 6. However, when the laser power further increases, as evident in Figure 5, the originally fine and uniform red regions gradually transform into large, randomly distributed red areas, leading to an increase in porosity on the sintered surface and a subsequent decrease in conductivity. This is because when the laser power exceeds a certain threshold, the surface temperature of the copper paste becomes excessively high, imparting more energy to the nanoparticles and intensifying the sintering process. This results in coarsening and fusion of particles, increasing their size and forming visibly grown pores. These large pores reduce film density and disrupt conductive channels within the material, leading to an increase in the resistivity of sintered lines.

#### 3.1.2. Effect of Scanning Speed on Sintered Circuit Lines

Laser sintering is a rapid heating process, where the sintering speed determines the sintering time, which in turn affects the amount of laser energy absorbed by the material. Therefore, the laser scanning speed is also a critical parameter influencing sintering quality. Under the conditions of a fixed laser power of 2 W and a single sintering pass, experiments were conducted by adjusting the laser scanning speed to 10 mm/s, 20 mm/s, 50 mm/s, 100 mm/s, 150 mm/s, and 200 mm/s. The surface micro topographies of sintered lines at different laser scanning speeds are shown in Figure 7. It can be seen from the SEM diagram that when the nano-copper paste is sintered at the speed of 10 mm/s, the copper nanoparticles on the surface of the sintering line have been sintered into blocks and completely sintered together. This is because when the laser scanning speed is slow, the sintering time is longer under the same laser power, and the sintering temperature is higher with the accumulation of heat. The copper nanoparticles melt and combine into larger particles, and form uneven gaps after cooling and solidification. As shown in Figure 8, the oxidation peak appears at this time, which may be due to the thermal decomposition of PVP covered by the surface of copper nanoparticles during the sintering process, and the exposure of copper nanoparticles to air leads to a slow sintering speed and longer exposure to air, resulting in more copper oxide; however, alcohol and acid after PVP decomposition are not enough to completely reduce the generated copper oxide to copper. The oxidation peak disappears when the speed increases, and this may be because, during the sintering process, PVP coated on the surface of copper nanoparticles encounters thermal decomposition, exposing copper nanoparticles to air, increasing the sintering speed, shortening the exposure time of copper nanoparticles to air, and reducing the generated copper oxide. The alcohol and acid after PVP decomposition will completely reduce the copper oxide to copper.

When the scanning speed increases to 50 mm/s, the intensity of sintering caused by heat accumulation decreases, leading to an improvement in sintering quality. The removal of organic material allows plastically deformed nano-copper to fill the gaps, resulting in a more densely sintered surface, as shown in Figure 7c.

As the speed further increases to 100 mm/s and 150 mm/s, the heating induced by the laser promotes the formation of necks, and a network-like structure begins to spread. However, despite these spreading effects dominating, a small portion of larger-diameter particles do not fully form sintering necks, as illustrated in Figure 7d,e.

When the speed reaches 200 mm/s, large particles are formed based on the migration and agglomeration of smaller particles. Figure 7f demonstrates particle growth occurring with coarsened particles covering the sintered line surface. There is no direct neck formation and densification from one particle’s surface to another; instead, coverage primarily consists of large particles.

The experiment also investigated the impact of laser scanning speed on the surface density and electrical properties of copper paste sintering. ImageJ software was employed to analyze the porosity of the sintered SEM images based on differences in brightness and darkness. Additionally, the electrical conductivity of the sintered lines at different scanning speeds was measured and calculated using the four-point probe method. Figure 9 illustrates the variations in porosity and electrical conductivity of the sintered lines with laser scanning speed, while keeping the laser power fixed at 2 W. It was observed that as scanning speed increased, the porosity of the sintered lines generally decreased, whereas there was a trend of an initially increasing and then decreasing relationship between electrical conductivity and scanning speed.

Upon analyzing the data in conjunction with the SEM images, it is evident that lower laser scanning speeds lead to a longer sintering time and intense sintering due to heat accumulation. This results in significant porosity within the copper film, disrupting the formation of conductive pathways and significantly impacting its electrical conductivity. Similar to the experimental results regarding power magnitude discussed in the previous section. When the scanning speed increases to 50 mm/s, it can be seen from Figure 9 that under this parameter, the porosity of the sintering line decreases the most and the densification degree is high. In addition to a few independent large-diameter copper nanoparticles, sheets of mesh interconnecting structures spread out and occupy a dominant position, the formation of these necks and grain growth form a direct conductive path, the contact effect becomes the main factor affecting the conductivity, and the more electronic paths, the better the conductivity.

As the scanning speed continues to increase, there is a slight decrease in the porosity of the sintered line surface, which remains relatively stable. However, the electrical performance of the sintered line decreases. Analysis of the SEM images of sintering reveals that excessively high scanning speeds reduce heat accumulation on the sintered line surface, leading to insufficient sintering and a decrease in electron paths. The inhibition of porosity increases due to the coverage of under-sintered particles, which is also observed. In particular, when the scanning speed increases to 200 mm/s, neck-like structures formed between small particles mainly exist in the form of points, and the surface is predominantly covered by larger nanoparticles with no obvious densification trend. Consequently, there is a decrease in electrical conductivity.

#### 3.1.3. Influence of Sintering Times on Sintering Lines

Due to the rapid heating and cooling process of laser sintering, a single sintering may not achieve the desired densification effect, especially with low laser energy density. Therefore, further study and analysis are needed to determine whether multiple sintering can improve the quality of sintering. This section investigates the impact of laser sintering times on the surface morphology and electrical conductivity of the sintered line. The surface morphology and electrical conductivity after multiple sintering are studied and analyzed under fixed laser power at 2 W with varying scanning speeds at 20 mm/s, 50 mm/s, 100 mm/s, and 150 mm/s.

The microstructure of the line after multiple sintering at different scanning speeds is shown in Figure 10. It can be observed from the microscopic topography that when the scanning speed is 20 mm/s, the surface topography formed on the sintering line becomes coarser with an increase in laser scanning times. This effect is especially pronounced when the laser sintering is continuous for four times, leading to significantly larger pores on the sintering surface compared to the first time. It is evident that at low scanning speeds, heat accumulation resulting from multiple scans promotes pore enlargement. When the speed is 50 mm/s, the laser-induced heating behavior drives the formation of the neck when the repetition is one to two times, and the sheet mesh structure begins to diffuse, which dominates the diffusion and increases the degree of densification. When the number of scans is increased by three to four times, the heat obtained by the nanoparticles accumulates, making the particles further melt and solidify. When the speed increases to 100 mm/s and 150 mm/s, after a single laser scan, necks form between small copper nanoparticles and become interconnected. However, many un-sintered particles remain on both the surface of the copper paste layer and within the film. With an increase in scanning times, although the network structure spreads out, an under-sintering phenomenon does not significantly improve, and densification remains inadequate.

The relationship between the porosity and conductivity of the sintering line and the sintering times at different speeds is illustrated in Figure 11. At a velocity of 20 mm/s, the porosity of the sintered surface increases from 14.61% for primary sintering to 18.42% for four sintering, accompanied by a slight decline in conductivity. Analysis of the corresponding SEM diagram in Figure 10 reveals that at low scanning speeds, heat accumulates with an increase in scanning times, causing further melting and re-solidification of originally formed particles, resulting in thick lines. The increase in particle size leads to increased porosity, with large pores disrupting the conductive channel of the material and leading to increased resistance. When the speed is 50 mm/s, the laser-induced heating behavior drives the formation of the neck at one to two repetitions, and the sheet mesh structure begins to diffuse, which dominates the diffusion and increases the degree of densification. At three to four repetitions, with the increase of scanning times, the nanoparticles gain heat and accumulate, making the particles further melt and solidify. The coarsening growth of the particles leads to an increase in porosity, which hinders the electronic transition and thus decreases the conductivity, resulting in an increase and then a decrease in the conductivity after two repetitions. When the speed is further increased to 100 mm/s, it can be observed from the Figure 11b data that at this point, the porosity of the sintered surface stabilizes at around 10%, with multiple scanning not significantly altering its value. Electrical conductivity fluctuates within a small range as scanning times increase; this may be attributed to stable porosity on the sintered surface due to high scanning speed reducing energy accumulation impact. When the speed is 150 mm/s, although the porosity of the sintering line increases when the laser scanning is repeated two to three times, the conductivity of the sintering line will slightly increase. The analysis of the corresponding microscopic morphology in Figure 10 shows that the surface porosity of the sintering line increases after multiple sintering at a high scanning speed. However, the heat accumulation caused by multiple sintering causes the neck interconnecting structure between small nanoparticles to form, increasing the conductive channel. Therefore, under constant laser power outputting conditions when laser scanning speed is low (resulting in energy accumulation caused by multiple scans), there will be a minor increase in the surface porosity of the sintering line. As the scanning speed increases, however, the influence on sintering results decreases due to reduced impact from energy accumulation during multiple scans. As shown in Figure 11b, with the increase of laser scanning repeats at different scanning speeds, heat accumulates on the sintered surface, the previously formed particles are further melted and resolidified, and the coarsening growth of particles reduces the densification degree. Therefore, the porosity of the sintering line increases with the increase of laser sintering repetition times.

### 3.2. Evolution Process of the Microstructure during Sintering of Nano-Copper Paste

Based on the analysis of the surface morphology of the sintered lines under various laser process parameters as outlined in Section 3.1, Figure 12 illustrates the evolution process of the microstructure during laser sintering of nano-copper paste. The sintering mechanism of nano-copper paste can be elucidated through the four steps depicted in Figure 12:

(1)Prior to sintering, the nano-copper particles are enveloped by an organic coating layer and exist independently of each other.(2)When a relatively low laser energy density is applied to the surface of the copper paste, the residual organic matter within the paste decomposes, causing the nano-copper particles to become closely arranged.(3)As the temperature increases, the nano-particles start to coalesce, with necks forming preferentially between smaller particles. This process is driven by surface diffusion in order to minimize the surface area for densification. Aggregation between particles leads to the formation of multiple particle clusters.(4)Under high energy density, an elevated temperature accelerates particle growth and plastic deformation becomes dominant. The deformed particles fill in pores, resulting in smaller pore sizes and increased density.

## 4. Conclusions

This paper primarily investigates the influence of laser process parameters, including laser power, scanning speed, sintering times, and variable parameters, on the surface morphology and electrical properties of sintered nano-copper paste. The experimental results show that the conductivity of the sintering line increases first and then decreases with the increase of the laser power and scanning speed. When the laser power is 2 W, the scanning speed is 50 mm/s, the number of sintering lines is double, the highest conductivity is achieved, and the best average conductivity is 3.46 × 10^6^ S/m. Furthermore, the microstructural evolution of nano-copper paste during laser sintering is summarized based on experimental observations. The main conclusions drawn are as follows:

(1)This study analyzes the effect of laser power on the sintered lines. It is found that the conductivity of the sintered lines exhibits a trend of first increasing and then decreasing with an increase in laser power. When the laser scanning speed is fixed at 50 mm/s, at low power levels (<1 W), organic matter in the copper paste is removed, and nanoparticles begin to make contact and arrange closely. After laser irradiation, nanoparticles (NPs) primarily exist in point contacts, indicating an initial stage of sintering. As power increases (≥1 W), nano-copper particles interconnect and form necks. With further power increases, sintering temperature rises leading to Ostwald ripening where necks form between particles enhancing densification of the sintered lines, improving conductivity. However, when the power exceeds the optimal sintering level, intense heat results in agglomerated particles melting and solidifying into blocks combined with thermal stress generated by high temperatures, which leads to increased spacing porosity, hindering electron transport and thereby increasing resistance.(2)The impact of laser scanning speed on the surface morphology and electrical properties of sintered lines is examined in this study. At a constant laser power of 2 W, the optimal conductivity is achieved at a scanning speed of 50 mm/s. An increase in scanning speed leads to decreased conductivity due to shorter sintering times, resulting in under-sintering of the copper paste layer. Higher scanning speeds have a relatively low impact on porosity due to inadequate powder coverage. Conversely, lower than optimal scanning speeds expose the nano-copper paste to high temperatures for an extended period, posing a risk of oxidation during sintering. Prolonged heat accumulation generates high temperatures, causing surface coarsening of the sintered lines.(3)This study also investigates the effect of laser sintering times on the surface morphology and electrical properties of sintered lines. With a fixed laser power of 2 W, multiple sintering cycles are analyzed at scanning speeds ranging from 20 mm/s to 200 mm/s. It is observed that at lower scanning speeds, multiple sintering cycles under the same power increase the risk of porosity and result in decreased conductivity. However, as the scanning speed increases (≥50 mm/s), multiple sintering cycles have minimal impact on porosity and conductivity without changing the power due to lower energy accumulation.(4)Finally, this study summarizes the microstructure evolution process of sintered copper nanoparticles based on experimental phenomena observed during testing.

## Figures and Tables

**Figure 1 nanomaterials-14-01426-f001:**
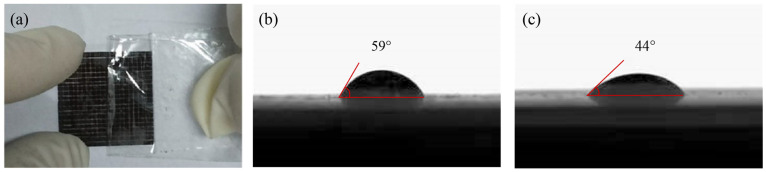
(**a**) 3M tape test. (**b**) Contact angle between nano-copper paste and substrate without adding PVP. (**c**) The contact angle between the nano-copper paste and the substrate after adding PVP.

**Figure 2 nanomaterials-14-01426-f002:**
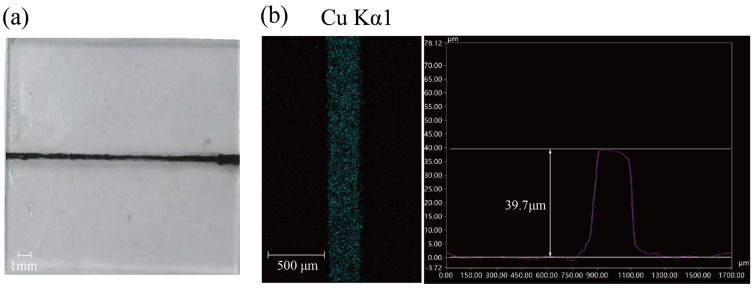
(**a**) Physical drawings of nanoscale copper paste coating wires; (**b**) 3D profile measurement results.

**Figure 3 nanomaterials-14-01426-f003:**
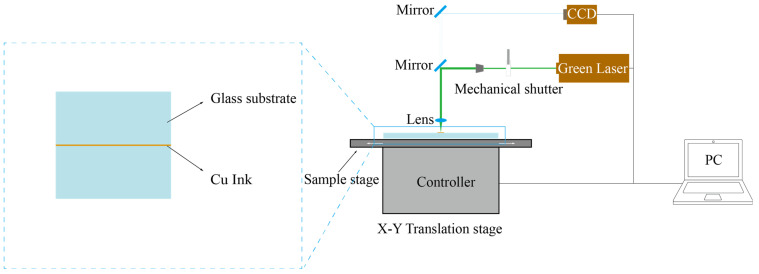
Diagram of experimental setup.

**Figure 4 nanomaterials-14-01426-f004:**
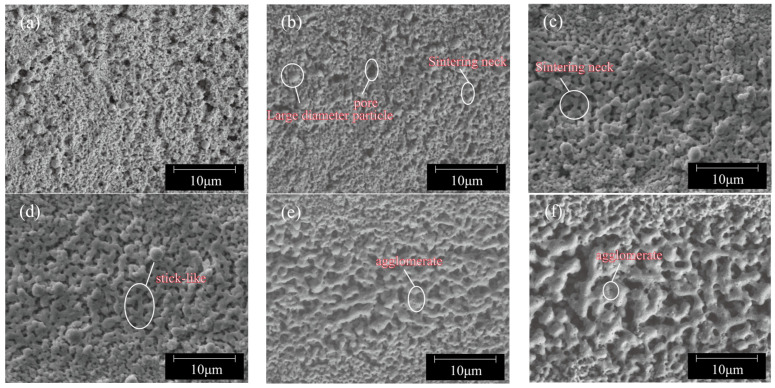
Surface morphologies of sintered lines under different laser powers: (**a**) 0.5 W; (**b**) 1 W; (**c**) 1.5 W; (**d**) 2 W; (**e**) 2.5 W; (**f**) 3 W.

**Figure 5 nanomaterials-14-01426-f005:**
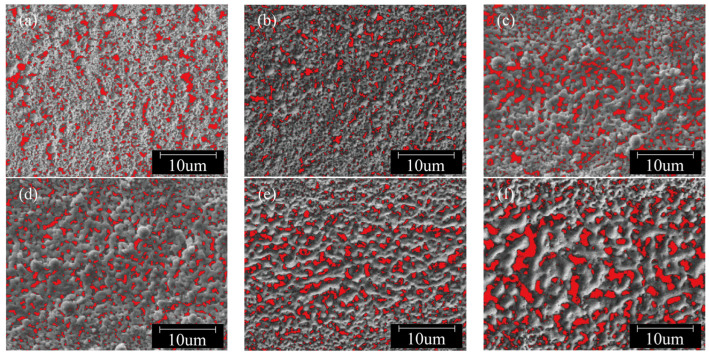
SEM images processed by ImageJ: (**a**) 0.5 W; (**b**) 1 W; (**c**) 1.5 W; (**d**) 2 W; (**e**) 2.5 W; (**f**) 3 W.

**Figure 6 nanomaterials-14-01426-f006:**
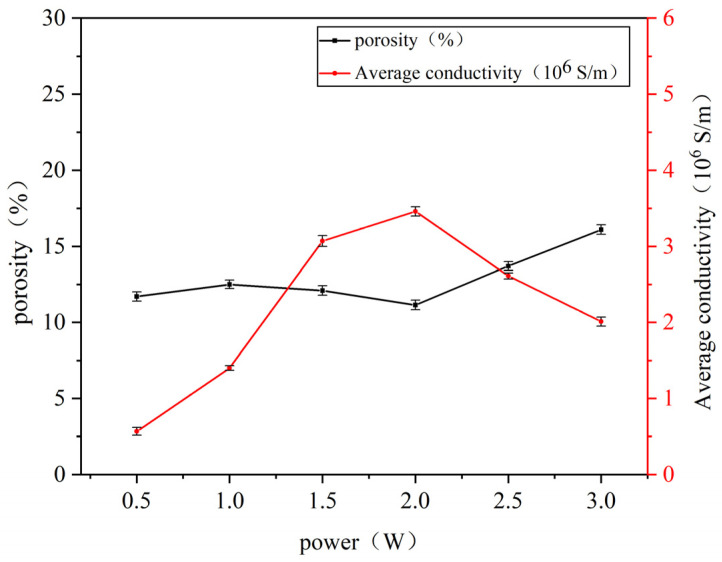
Relationship between porosity, conductivity of sintered lines, and laser power.

**Figure 7 nanomaterials-14-01426-f007:**
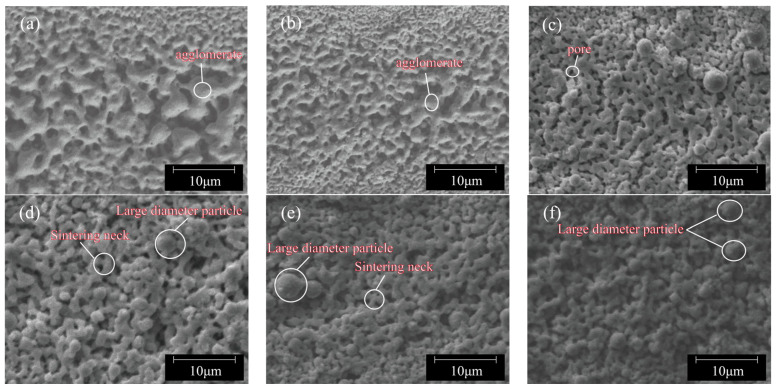
Surface morphologies of sintered lines at different scanning speeds: (**a**) 10 mm/s; (**b**) 20 mm/s; (**c**) 50 mm/s; (**d**) 100 mm/s; (**e**) 150 mm/s; (**f**) 200 mm/s.

**Figure 8 nanomaterials-14-01426-f008:**
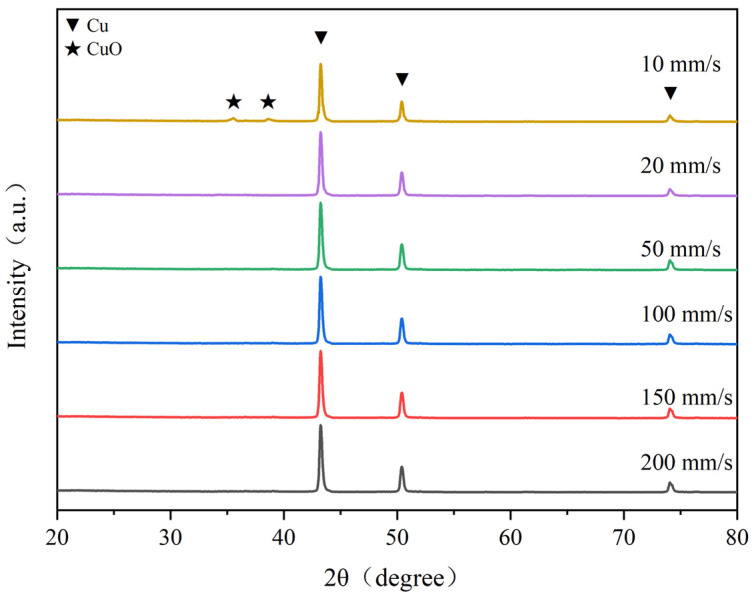
XRD patterns of sintered samples at different scanning speeds.

**Figure 9 nanomaterials-14-01426-f009:**
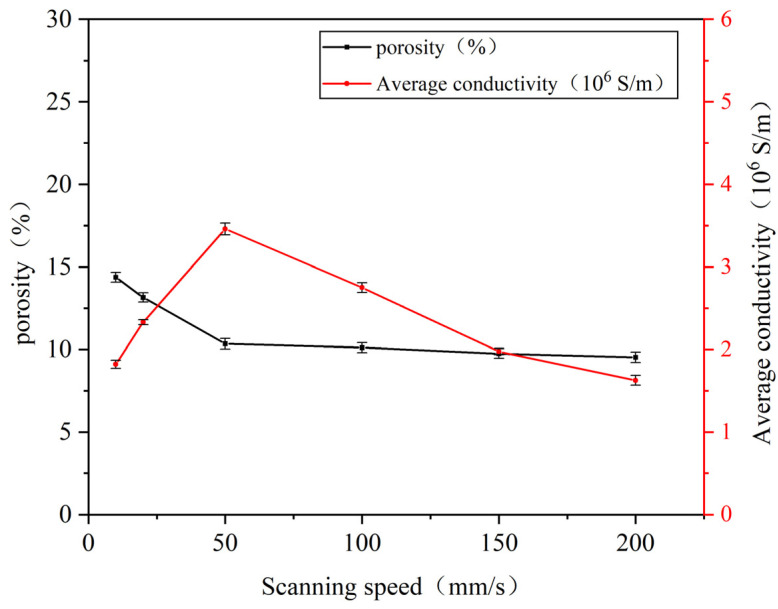
Relationship between porosity and electrical conductivity of sintered lines versus laser scanning speed.

**Figure 10 nanomaterials-14-01426-f010:**
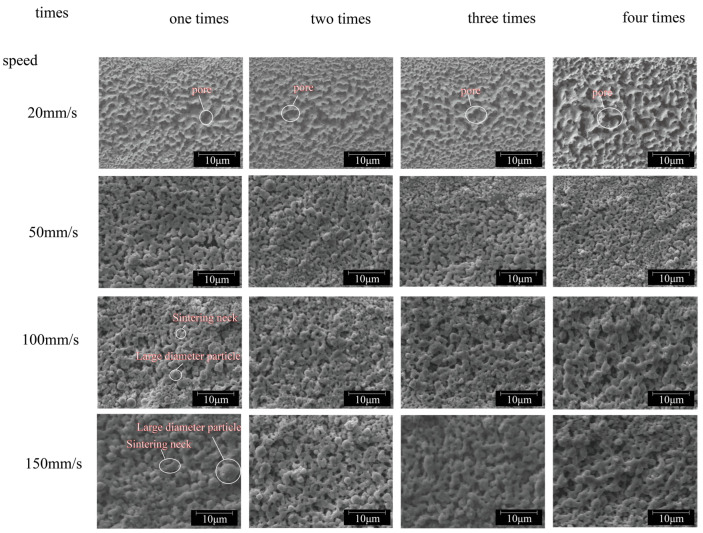
Microscopic morphology of multiple sintered lines at different scanning speeds.

**Figure 11 nanomaterials-14-01426-f011:**
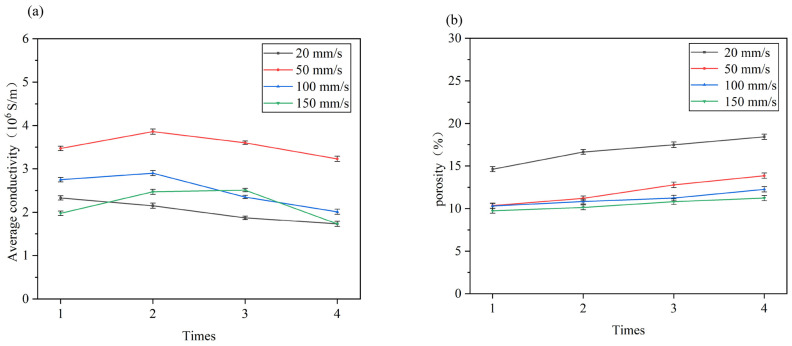
At different scanning speeds, (**a**) the relationship between electrical conductivity and sintering times. (**b**) The relationship between porosity and sintering times.

**Figure 12 nanomaterials-14-01426-f012:**
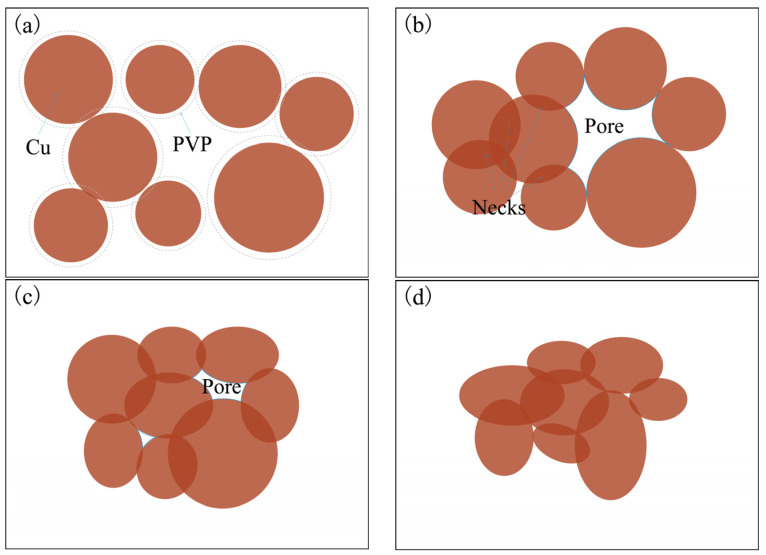
Schematic diagram of the sintering mechanism of copper powder. (**a**) Un-sintered; (**b**) organic decomposition and particle contact; (**c**) neck formation and diffusion; (**d**) densification and growth.

## Data Availability

The research data in question pertain to sensitive company information and are not suitable for public disclosure.

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
