# Peer review of "Study on the Technology and Properties of Green Laser Sintering Nano-Copper Paste Ink"

_nanomaterials, 2024, doi:10.3390/nano14171426_

Round 1

Reviewer 1 Report

Comments and Suggestions for Authors

This paper is generally well written and clear. It describes the study of the properties of laser sintered printed lines made of nano-copper paste inks, and is mainly focused on the influence of the different conditions of sintering, namely: the laser power, the scanning speed, the number of sintering passes. Studied properties are the porosity of the resulting lines and their conductivity. Results are analyzed, with an effort made to interpret them.

However, further discussion of the link between porosity and conductivity would be appreciated. Clearly, two porous states described by the same porosity value do not lead to the same conductivity. This raises the question of whether porosity measurement using 2D SEM observation is sufficient to explain the differences observed.

Temperature of the sintered lines is often mentioned as a key parameter, but is never measured nor evaluated. Sintering temperature is compared to melting point of copper nanoparticles (in § 3.1.1) but no value is mentioned.

On the graphs, it is preferable to show the uncertainties (Figures 5, 7, 8, 10).

In the conclusion, it would have been wise to summarize the best settings (power, speed and number) for laser sintering to achieve the highest conductivity.

Miscellaneous comments:

-       Very few details are mentioned about the properties of the copper paste ink (viscosity? surface tension?) as well as the printing conditions (§2.1.and 2.2).

-       We suggest to enlarge figure 1 for better legibility.

-       In § 2.3, the sentence “The suitability of different laser parameters for the sintering process was compared” is incomplete: compared to what?

-       In § 2.3: “Figure 3” is “Figure 2” indeed.

-       In § 2.4: in equation (1) what is “m”?

-       The last sentence in §2.4 is redundant.

-       §3.1.1: “Influence of laser power on sintered line” instead of “sintering lines”.

-       § 3.1.3: in the title of the § and in the text, the term “frequency” is inappropriate; is it the “number” of sintering passes? Later in the §, “times” is mentioned, and “cycles” is used in the conclusion. Please clarify.

-       § 3.2 is a description of the evolution of the microstructure of the nano-copper paste during sintering. This process should be compared to similar situations, with other sintering method, for instance.

-       There are some redundancies in the beginning of the conclusion.

Comments on the Quality of English Language

Please check some of the terms mentioned in the previous comment.

The English is mostly OK.

Reviewer 2 Report

Comments and Suggestions for Authors

This paper deals with the effect of green laser irradiation on the sintering characteristics of nano-sized copper particles when applied to copper ink paste printed on glass substrates to fabricate electronic circuit prototypes. The following corrections and improvements are required before publication:

- The sintering behavior of nano-sized metal particles is expected to vary depending on the atmosphere during laser sintering. In the experimental method, specify the atmosphere used for this laser sintering.

- Nano-sized copper particles are easily oxidized in air and are highly flammable, which increases the risk of fire. As shown in the XRD results in Figure 7, nano-sized copper particles are easily oxidized, so laser sintering is required in an inert gas or vacuum atmosphere rather than an oxidizing atmosphere. Therefore, it is necessary to present the factors affecting sintering in an inert gas or vacuum atmosphere with ventilation rather than an oxidizing atmosphere.

- In Figure 8, the highest electrical conductivity is observed at 50 mm/s. However, many large spherical particles are also observed at 50 mm/s in the SEM image in Figure 6c. Explain why the electrical conductivity increases the most despite the presence of a significant number of large spherical particles.

- It is unclear why the porosity of the sintering line increases with increasing number of laser sintering repetitions at various scan speeds in Figure 10b.

- Also explain why the electrical conductivity increases in Figure 10a when the laser scan is repeated 2-3 times despite the increasing porosity of the sintering line.

- Page 9, lines 9-10: The SEM image caption in Figure 9 states that there is no significant change in the microstructure with increasing number of laser sintering repetitions at 50 mm/s. Explain why the porosity increases despite no change in the microstructure, and why the electrical conductivity increases and then decreases after two repetitions.

Numerous typing errors, including spacing, were found in the text and figure captions and need to be corrected. Example:

- Page 2, Line 7: "This study utilizes a 532nm continuous green laser to laser-sinter 40um wide nanometer copper ink printed on glass substrate in order to investigate the feasibility of using laser-sintered nanometer copper paste for forming electronic circuits" should be corrected to " This study investigates the feasibility of using laser-sintered nanometer copper paste to form electronic circuits by utilizing a 532 nm continuous green laser to sinter 40 μm wide nanometer copper ink printed on a glass substrate".

- Frequency units should be added in Figure 9.

- Unlike other figure captions, Figures 1, 2, and 10 capitalize only the first sentence. Please style all figure captions in a consistent manner.

Reviewer 3 Report

Comments and Suggestions for Authors

Nanometer copper ink printed on a glass substrate is laser-sintered using a 532nm green laser, and the feasibility is studied in relation to laser power, scanning speed, and frequency.

Paragraph 2.1 Materials - Explain the origin of the copper nano-ink.

Paragraph 2.2 Sample Preparation - It might be relevant to mention the equipment used for printing.

Figure 1(a) - Introduce a scale bar for the image of the printed line.

Paragraph 2.4 Surface Characterization - Include the name of the SEM and XRD equipment.

Equation 1 - The "m"  is not explained in the text.

Figures 3, 4, 6, 9 - The SEM microscopy images are processed; it might be necessary to retain the original bar with the SEM acquisition parameters.

The results of each analyzed parameter related to the material synthesis are well-analyzed.

The conclusions are well-organized.

Round 2

Reviewer 1 Report

Comments and Suggestions for Authors

Thank you for taking into account the remarks of the first review.

The explanations and clarifications provided in the new version are convincing.